# A Machine Learning Approach to Robot Localization Using Fiducial Markers in RobotAtFactory 4.0 Competition

**DOI:** 10.3390/s23063128

**Published:** 2023-03-15

**Authors:** Luan C. Klein, João Braun, João Mendes, Vítor H. Pinto, Felipe N. Martins, Andre Schneider de Oliveira, Heinrich Wörtche, Paulo Costa, José Lima

**Affiliations:** 1Department of Electronics (DAELN), Universidade Tecnológica Federal do Paraná (UTFPR), Curitiba 80230-901, Brazil; luanklein@alunos.utfpr.edu.br (L.C.K.); andreoliveira@utfpr.edu.br (A.S.d.O.); 2Research Center in Digitalization and Intelligent Robotics (CeDRI), Instituto Politécnico de Bragança, 5300-253 Bragança, Portugal; jbneto@ipb.pt (J.B.); joao.cmendes@ipb.pt (J.M.); jllima@ipb.pt (J.L.); 3Laboratório para a Sustentabilidade e Tecnologia em Regiões de Montanha (SusTEC), Instituto Politécnico de Bragança, 5300-253 Bragança, Portugal; 4Faculty of Engineering, University of Porto (FEUP), Rua Dr. Roberto Frias, 4200-465 Porto, Portugal; vitorpinto@fe.up.pt (V.H.P.); paco@fe.up.pt (P.C.); 5INESC Technology and Science, 4200-465 Porto, Portugal; 6ALGORITMI Center, University of Minho, 4710-057 Braga, Portugal; 7SYSTEC (DIGI2)—Research Center for Systems and Technologies (Digital and Intelligent Industry Lab), 4200-465 Porto, Portugal; 8Sensors and Smart Systems Group, Institute of Engineering, Hanze University of Applied Sciences, 9747 AS Groningen, The Netherlands; h.j.wortche@pl.hanze.nl; 9Department of Electrical Engineering, Eindhoven University of Technology, 5600 MB Eindhoven, The Netherlands

**Keywords:** indoor localization, machine learning, fiducial markers, Industry 4.0, robotics competitions

## Abstract

Localization is a crucial skill in mobile robotics because the robot needs to make reasonable navigation decisions to complete its mission. Many approaches exist to implement localization, but artificial intelligence can be an interesting alternative to traditional localization techniques based on model calculations. This work proposes a machine learning approach to solve the localization problem in the RobotAtFactory 4.0 competition. The idea is to obtain the relative pose of an onboard camera with respect to fiducial markers (ArUcos) and then estimate the robot pose with machine learning. The approaches were validated in a simulation. Several algorithms were tested, and the best results were obtained by using Random Forest Regressor, with an error on the millimeter scale. The proposed solution presents results as high as the analytical approach for solving the localization problem in the RobotAtFactory 4.0 scenario, with the advantage of not requiring explicit knowledge of the exact positions of the fiducial markers, as in the analytical approach.

## 1. Introduction

The problem of localization in robotics is related to the question, “Where am I?”, as asked by a mobile robot. Localization is one of the most fundamental competencies required by an Autonomous Mobile Robot (AMR) as the knowledge of the robot’s location is an essential pre-requisite for making decisions about future actions [1]. Localization can be understood as estimating the pose (position and orientation) of a mobile system relative to a reference frame, usually the global reference frame. It is essential to note that one of the most famous localization systems, the Global Positioning System (GPS), may not be available in indoor environments because it has the limitation of the satellite signals being blocked/attenuated by structures such as walls and ceilings [2]. In this way, other alternatives are necessary to solve this problem, especially when robots are operating indoors.

In this context, several approaches have been developed over time, using sensor data and several algorithms. The use of artificial intelligence (AI), especially machine learning (ML), is becoming a common approach to this problem [3]. Furthermore, there are many robot competitions where localization is a fundamental principle. One example of this type of competition is RobotAtFactory 4.0 (https://www.festivalnacionalrobotica.pt/2023/robotfactory-4-0/, accessed on 10 December 2022), where the AMR has to move boxes from one place to another in the shortest possible time, without external communication. In this case, the AMR can use whichever means is necessary to locate itself as long as it complies with the rules, and one possibility is to use markers placed in the environment (ArUco markers). Figure 1 shows a top view of the field. The reference frame is located at the center of the field, where the green arrow indicates the *y* axis, the red arrow indicates the *x* axis and the numbers indicate the IDs of each ArUco marker. The black numbers indicate the tag’s ID on the floor, while the blue numbers indicate the tag’s ID on the walls.

One of the current approaches to solve the localization problem at the RobotAtFactory 4.0 competition is using the stored data of the ArUcos pose and the relative position of the ArUcos with the camera reference frame, using geometry, to estimate the robot pose [4]. Each identified ArUco has an estimated pose, and, with some filters (such as Kalman and Mahalanobis filters), it is possible to aggregate these poses and generate a good estimate of the robot pose. In this case, the exact pose of the ArUcos needs to be known and small errors in the data can have a strong impact on the estimates of the robot’s position.

In this paper, we propose to solve the localization problem using ML to estimate the robot’s pose. Two similar approaches are tested. In both, the robot’s pose is estimated using the ArUcos identified by the robot’s onboard camera. The difference is that in approach 1, the robot’s pose is estimated using all ArUcos identified in one image frame at once, while, in approach 2, an estimated pose for each ArUco is calculated. The goal of approach 1 is to solve the robot localization problem at once, while the goal of approach 2 is to provide estimates to feed filters (such as Extended Kalman Filter), which will work as a complete localization system. The focus of this work is to first validate the application of ML to solve the localization problem in a simulated scenario.

The work is divided into four more sections: Section 2 presents the state of the art in localization; Section 3 presents theoretical concepts and explains how the work was developed, namely the data collection, the preprocessing of data and the models used; Section 4 presents the results and the discussion. Finally, Section 5 presents the conclusions of this work and notes for future works.

## 2. State of the Art

The study of localization is an area that has received a lot of attention from the community in recent years. In outdoor scenarios, the most common approach is to use GPS. However, this approach fails indoors because of physical limitations, consequently generating a demand for others solutions. Thus, in this way, different approaches have been proposed by several authors to solve indoor localization. Furthermore, it is important to note that in addition to estimating the pose, it is necessary to know the uncertainty associated with the estimate; there can be catastrophic consequences due to decisions made on the basis of pose estimates that are assumed to be perfect [1].

One of the most famous approaches is the Kalman Filter, first proposed in [5]. It is a mathematical approach that uses measurements over time (containing noise) to generate values that approximate the actual values. It estimates a process by using a form of feedback control: the filter estimates the process state at a given moment and then obtains feedback in the form of (noisy) measurements [6]. Moreover, a Gaussian distribution is used in this method to inject noisy data into the estimates [7,8]. The Kalman Filter is linear and used in linear situations but, actually, the majority of situations are not linear. However, some variants of the Kalman Filter can be used in nonlinear situations, such as the Extended Kalman Filter (EKF) [6].

Another famous approach is Markov localization (also known as grid localization). The main idea of this approach is to discretize the space of possible robot locations and manipulate discrete probability distributions [1]. Markov localization is a probabilistic algorithm. Instead of maintaining a single hypothesis as to where in the world a robot might be, it maintains a probability distribution over the space of all such hypotheses [9]. For this, the approach uses a histogram for all degrees of freedom.

A further interesting approach is Monte Carlo localization, first introduced in [10] (as the bootstrap filter) and further explained in [11]. This algorithm is a version of the particle filter [12,13]. Multiple samples (particles) are used to represent a hypothesis of the interest variable, i.e., the robot localization. Each hypothesis is associated with a weight, representing the likelihood that the hypothetical estimate is the true value. A pose estimate can be obtained by the weighted sum of all samples.

Several other approaches have been developed in the last few years, such as the Perfect Match approach, first proposed by [14], and developed initially for robot localization in the Robocup Midsize league. Another approach is the Iterative Closest Point (ICP) algorithm, introduced by [15], which is a commonly used map-matching method. This approach tries to minimize the Euclidean distance between the input data and a reference model (in the localization problem, it corresponds to the sensor data and the map of the environment) [16]. The Normal Distributions Transform (NDT) approach, first introduced by [17], is a method for 2D scan registration and it was later extended to 3D [18,19]. This approach is a map-matching-based algorithm, and it creates a smooth surface representation of the environment that is modeled by a set of local probability density functions. Another interesting approach to solving the indoor localization problem is using pulse-echo ultrasonic signals [20].

The presented approaches can be used in the RobotAtFactory 4.0 competition, but some of these, such as the Extend Kalman Filter, can perform better when fed with good pose estimates. Furthermore, some of them can be overly complex. In this way, other solutions can be interesting, such as the use of landmarks in the environment. The use of fiducial markers in localization is a very common approach to this problem [21]. The use of fiducial markers in a Simultaneous Localization and Mapping (SLAM) problem with other localization techniques (e.g., Monte Carlo) was also explored [22]. As previously mentioned, in the RobotAtFactory 4.0 competition, an analytical approach using fiducial markers has been proposed by [4].

Recently, with the advancement of AI, also some AI-based algorithms have been developed. For instance, the Convolutional Neural Network (CNN) was used to help locate the robot using images from a robot camera and other sensors [23,24]. Interesting related work is presented in [25], where the authors used images for localization in a scenario with 6-DoF (Degrees of Freedom). The authors used the transfer learning of GoogLeNet [26] and obtained good results in outdoor and indoor scenarios.

The use of ML techniques for indoor positioning is an interesting approach that has become increasingly prominent. Survey research on this topic was performed in 2020 and shows that several approaches, such as Random Forest (RF), Support Vector Machines (SVM), K-Nearest Neighbors (KNN), Artificial Neural Networks (ANN), and others (such as a combination of these approaches), have been used to extract and select features and perform regression in a localization context [3]. This survey shows that this area is in its infancy, and several issues need further investigation.

To the best of our knowledge, no work focused on the use of fiducial markers and ML for localization was published up to now. The closest approach found was using ML to improve the quality of detection of the fiducial markers [27]. The objective of the present work is to explore this promising approach.

## 3. Project Description

This section first gives a brief explanation of machine learning concepts and fiducial markers. Then, it explains the robot architecture and the simulator used in this work. It concludes with an exploration of the data process, including the data collection, preprocessing, models, and then evaluation.

### 3.1. Theoretical Background

Machine learning, by definition, is a computer technique that, from a collection of data, learns without explicit instructions and draws inferences from patterns in the data. This work, based on this concept, will apply supervised learning [7]. The input of the present system is images (which will be processed) and the output is the robot’s pose. The approaches used in this work are presented below.

**Random Forest** (RF) is an ensemble learning method that consists of a random combination of multiple decision trees. This method is a common approach to classification problems but can be used for regression as well [28].**K-Nearest Neighbor Regression** (KNR) is a regression version of the K-Nearest Neighbor (KNN) algorithm. This is one of the simplest and best-known non-parametric methods [29]. The main idea is to predict the target according to a local interpolation of the targets of the nearest neighbors in the training set (https://scikit-learn.org/stable/modules/generated/sklearn.neighbors.KNeighborsRegressor.html, accessed on 7 December 2022).**Multilayer Perceptron** (MLP) is a fully connected Artificial Neural Network (ANN) and was presented by [30]. It is composed of artificial neurons, whose model was proposed by [31]. MLP consists of at least three main layers: an input layer, a hidden layer (one or more), and an output layer. MLP uses the back-propagation learning technique for training [32].**Gradient Boosting** (GB) is another ensemble technique and was presented by [33]. The main idea is to build the model in a sequential way. The basic idea is to add a new model in each iteration to try to improve the previous models [34].

### 3.2. Fiducial Markers

Fiducial markers in their general form are objects used to provide a point of reference or a measurement in an image [21]. There are several types of fiducial makers, such as ARTag, AprilTag, ArUco, and STag [21]. These approaches are different from each other, such as the way in which each of them is built. Figure 2 presents a comparison between these four types of fiducial markers. For further information and a comprehensive discussion of specific features, the reader is referred to [35]. One of the most common types is the binary square fiducial marker. An example of this type is ArUco, which is present in the OpenCV library (https://docs.opencv.org/4.x/d5/dae/tutorial_aruco_detection.html, accessed on 30 November 2022) and was developed by [36]. In this work, the ArUco will be used because this is the type used in the RobotAtFactory 4.0 competition.

### 3.3. Scenario Context

There are several robot competitions and they are an interesting way to encourage and attract students to the technological field, introducing new technologies and tools [37,38]. Furthermore, with this stimulus, students can develop their critical thinking and problem solving, thus developing soft and hard skills. The focus of this work is to develop localization approaches in the RobotAtFactory 4.0 competition scenario.

In this competition, the robot has to move boxes from the entrance and exit of the warehouse, called incoming and outgoing warehouses, respectively. The main goal is to move as many boxes as possible in the shortest time. The idea of this competition is to simulate a real scenario in automated factories, such as a warehouse, in which some material needs to be passed through several processes inside the environment. The warehouse is part of an automatized chain, so automation with robots can replace human workers and, in turn, improve efficiency and human safety. In this context, the Industry 4.0 theme emerged and has gained more attention and importance [39,40].

#### 3.3.1. Robot

The robot used in the RobotAtFactory 4.0 competition is subjected to several rules, such as fitting into a cube of 30 × 30 × 30 cm, and it must be completely autonomous. It cannot establish any form of communication with any external system that is not explicitly provided by the organization.

Figure 3 shows the main components of the robot. The Raspberry Pi deals with the high-level control of the robot (controlling, for example, the RGB camera, and the localization, navigation, and decision making) while the Arduino Uno manages the low-level control of the robot (e.g., motors and encoders) [41].

#### 3.3.2. Realistic Simulator

In the present work, only data from a simulated scenario were used, which offer quick implementation and validation of the approaches proposed. The simulator used in the RobotAtFactory 4.0 competition is the SimTwo simulator (the simulator has already been developed and it is provided by the competition organization at https://github.com/P33a/SimTwo, accessed on 25 November 2022), which works with rigid-body dynamics, interactions, and constraints [42]. Figure 4 presents an image of the simulator that displays a virtual representation of the official RobotAtFactory 4.0 competition field. The simulated scenario follows the specifications of the official competition rules.

In the simulator, the user can interact with several windows. For instance, there is a window with an XML file editor, which enables the user to change some definitions regarding the environment and the robot. Furthermore, the simulator has other features, such as the code editor (in Pascal language) that allows the developer to program it—for example, building an algorithm to define the robot’s route [41]. More details about the simulator can be found in [42].

### 3.4. Methodology

A sequence of steps was followed to obtain the results and conduct the analysis. Figure 5 presents an overview of the process flow. The first step is data collection, which means collecting images from the robot’s camera. The second is the data preprocessing, which includes activities such as data selection, ArUco identification, data preparation, and the baseline definition. The third step is focused on the models. Two main approaches were used here. In the first approach, a model is created that receives a matrix with information about the relative pose of all tags identified in an image and that returns only one estimated pose per image. In the second approach, a model is created that receives only the information about one relative pose of one tag per time and that returns the estimated pose according to only this tag (this implies that an image with X tags identified will have X different position estimates). The main goal is, using the first model, to validate the quality pose estimation using ML, and, with the second model, to compare the predictions from ML against the analytical approach.

#### 3.4.1. Data Collection

To collect the data, two adjustments had to be made in the scenario to avoid problems that were not relevant to validating the ideas proposed here. First, the illumination was replaced to try to avoid dark parts on the field, and, second, the color of the walls was changed to white (because the library used to identify them works better when the edges are white). Figure 6 presents an example of an image from the simulator. Each image is associated with a global pose {*x, y, θ*} (where *x* and *y* are in meters and θ is the orientation in degrees). For example, the image in Figure 6 corresponds to the pose {−0.461,0.240,2.05}.

To collect the data, the field was discretized into a grid with each square having a side with a size equal to 1 cm. With this, the robot was positioned in all positions, i.e., without obstacles, on this grid. In each position, the robot takes 63 images whilst performing/making a 360° turn in order to create a database for ML training. This way, 347.403 images have been collected.

Furthermore, for a limited part of the field, four more datasets were generated, collecting images with a different grid resolution and, consequently, collecting different amounts of images. The part of the field used was 10 × 10 cm in the center of the scenario, presented in Figure 7. The grid resolution sizes used were 10 mm, 5 mm, 2.5 mm, and 1 mm. Finally, another small dataset with around 500 images was collected using a random route through the whole field, thus yielding 6 data collections in total.

#### 3.4.2. Data Preprocessing

The first preprocessing activity was to remove images that may be repeated or contain incorrect values due to delays in the simulator (because of computational limitations, some images can be saved twice or more, where the subsequent images are taken out of sync with respect to the pose of the camera). After this, the data are ready to be processed to create the datasets. In this context, it is important to highlight that only images that contain at least one ArUco were used. This is important to avoid problems with “ambiguous images”. For instance, Figure 8 is an ambiguous image because it can be associated with any corner of the field. This phenomenon occurs mainly on the border of the field, when the robot has the camera focused outside of the field. This treatment is important because these images do not contribute to training or model evaluation. It is important to emphasize that this problem will be solved in the future using a complete localization system that uses odometry and sensor fusion to fill the gaps in the proposed model.

After this, the next step is to process images and prepare the data for the model. To do this, the images are processed using the OpenCV library (https://docs.opencv.org/4.x/d5/dae/tutorial_aruco_detection.html, accessed on 30 November 2022) to identify the ArUcos (the version used was 4.6.0) and to estimate the pose of each ArUco relative to the camera’s reference frame. In this estimation, the OpenCV library returns arrays with the position and the orientation of the marker to the camera. These arrays are called *tvec* and *rvec*, respectively. Both arrays contain 3 elements, where each element corresponds to one axis (*x, y, z*). Figure 9 displays the detected markers alongside their reference frames, and their corresponding pose arrays are shown in Table 1. More details about the ArUco identification can be seen in [4].

Finally, it is possible to aggregate the data and create the datasets to train and test the models. Since this work has two different goals, some different datasets were created. Considering the data collected, seven datasets were created. These were as follows.

**Dataset A**—*Using images from the whole field and combining the observations’ attributes in a matrix format*: The images used in this dataset were collected in the whole field considering a grid with a 1 cm resolution. In this, each observation in the dataset represents one image and has only one feature and 3 targets. The feature is a matrix with 49 lines (where each represents one ArUco) and 7 columns (where each represents the *tvec* and *rvec* array data and the last column serves to identify whether the marker was detected or not). In the cases in which an image does not contain a specific ArUco, the corresponding line in the matrix will be 0 (indicating the absence of the ArUco in that image). The target variables are *x, y* and θ.**Datasets B**—*Using images from the limited part of the field with different grid resolutions and combining the observations’ attributes in a matrix format*: The images used in these datasets were collected in the center of the field (Figure 7) considering different grid resolutions. Four resolutions were used and four datasets were created: **B1** with 10 mm, **B2** with 5 mm, **B3** with 2.5 mm, and **B4** with 1 mm. The method to create each dataset was the same as presented for Dataset A.**Dataset C**—*Using images from the whole field in an ArUco’s array*: The images used in this dataset were collected in the whole field considering a grid with a 1 cm resolution. Instead of each observation representing an image, now, each one represents one detected ArUco. Thus, each observation is composed of seven features: the tag’s id, *rvecs* and *tvecs*; and three targets, *x, y* and θ. Thus, in this dataset, the information about the image is not relevant anymore, since the focus is on the relative pose of the ArUcos.**Dataset D**—*Using images from a random route in an ArUco’s array*: The images used in this dataset were collected in the whole field considering a random path. The method to create the dataset was the same as presented for Dataset C.

Finally, to create a base comparison parameter for the models that will predict the pose, it is necessary to define a **baseline**. This is the most basic possible model. In a simple way, this “model” calculates the average of the training set (for the three target values: *x, y* and θ) and uses it as the prediction for all cases. For example, let us suppose that the targets in the training set are T = {1, 2, 3, 4, 5} and the targets in the test set are Ts = {1, 2, 6}. The average of the training set is 3, so all predictions for the test will be 3.

#### 3.4.3. Processing and Model Definition

Three independent models were created, one for each variable (*x*, *y* and θ). Four algorithms were used: Random Forest Regression, KNR, MLP, and Gradient Boosting. The library used to implement these models was *scikit-learn* (https://scikit-learn.org/stable/, accessed on 19 December 2022), and the models were *RandomForestRegressor*, *MLPRegressor*, *KNeighborsRegressor*, and *GradientBoostingRegressor*. The first process to be executed was training each one of these algorithms using dataset A. To do this, the dataset was separated into two parts: 85% to train and 15% to validate. After the training, using the error measurements, which will be explained in Section 3.4.4, the models were evaluated. Figure 10 presents the described process. The model parameters used are the default parameters defined in Scikit-learn.

After this, the algorithm that presents the best results was trained using the four smallest datasets: B1, B2, B3, and B4. Again, the datasets’ split was 85% to train and 15% to test. Finally, using again the algorithm that presented the best results in the first process, the model was retrained using dataset C (built with the data from the whole field), but now using 100% of it to train. This model will be used to compare with the analytical approach using dataset D (built with the random route through the field). It is important to highlight that, in this part, only ArUcos on the floor are used (this limitation is used to ensure the possibility of comparison with the approach used by [4]).

Finally, it is important to highlight that the results in this work were obtained using a CPU, with an *AMD* (Founded: 1 May 1969, Sunnyvale, California, United States) *EPYC 7351 16-Core Processor (2.40GHz)* and 32 GB of RAM. No GPU was used for processing. The programming language used was Python (Downloaded from https://www.python.org/, accessed on 25 November 2022) 3.10.7, which is open-source. The following open-source packages were also used, all installed using the Python Package Index (PyPI) (https://pypi.org/, accessed on 25 November 2022): Pandas 1.5.0, OpenCV 4.6.0, and Scikit-learn 1.1.2.

#### 3.4.4. Model Evaluation

To evaluate the quality of the models, four metrics were used: Mean Absolute Error (MAE), Root Mean Squared Error (RMSE), Normalized Root Mean Squared Error (NRMSE), and $R^{2}$ [43]. These metrics are given by the following equations:(1)$$\mathrm{MAE}=\frac{1}{n}\sum_{i=1}^{n}\left|y_{i}-{\hat{y}}_{i}\right|,$$
(2)$$\mathrm{RMSE}=\sqrt{\frac{1}{n}\sum_{i=1}^{n}{\left(y_{i}-{\hat{y}}_{i}\right)}^{2}},$$
(3)$$\mathrm{NRMSE}=\frac{\mathrm{RMSE}}{y_{max}-y_{min}},$$
(4)$$R^{2}=1-\frac{\sum_{i=1}^{n}{\left(y_{i}-{\hat{y}}_{i}\right)}^{2}}{\sum_{i=1}^{n}{\left(y_{i}-{\bar{y}}_{i}\right)}^{2}},$$
where $y_{i}$ represents the true value and ${\hat{y}}_{i}$ represents the predicted value for the instance *i*; $y_{max}$ and $y_{min}$ represent the max and min values, respectively, of the observations, while $\bar{y}$ represents the mean. The best value possible for MAE and RMSE is 0, and the worst value is +*∞*, while for $R^{2}$, the range is (−∞, 1], where −∞ is the worst possible value and 1 is the best.

## 4. Results and Discussion

The first result to be shown is the comparison between the different ML approaches, considering the whole scenario, such results are presented in Table 2. The first approach presented is the baseline and next the ML approaches are presented. The sections MAE and RMSE indicate their respective errors, with *x* and *y* in meters and θ in degrees. The section NRMSE indicates the normalized RMSE, enabling the comparison between *x* and *y* and θ. The $R^{2}$ presents the coefficient of determination for each target value on each approach. The last two sections present the average time to train the models and the response time for each image.

To better illustrate the results, Figure 11 shows a graphical comparison between the approaches. The left graph presents the MAE error for axes *x* and *y* in meters, while the right graph presents the MAE error for the θ, in degrees.

The second result is a comparison between different sizes of grid resolutions during the data collection. In previous results, the grid resolution was 10 mm and the complete dataset for the entire field was used. Now, only a specific part of the scenario is used, but the methodology to train and validate the model is the same. The results are presented in Table 3, where, in addition to the quality metrics, the number of collected images is also shown.

Figure 12 presents a graphical depiction of the behavior of the MAE for the decreases in the grid’s resolution. The MAE for the *x* and *y* axes is the same and is in centimeters, presented by the left curve on the graphic. The MAE for θ is in degrees, shown by the curve on the right.

The third and final result that will be presented is the comparison between the ML approach (specifically the Random Forest Regressor) and the analytical approach prediction, considering the whole scenario. The metrics presented before were calculated for both approaches, and the results are shown in Table 4. It is important to note that to calculate the percentage relative error, the predictions where the ground truth value was 0 were discarded. This was necessary because of a possible division by 0 on the percentage relative error function, where the denominator is the ground truth value.

Table 2 presents four approaches to solving the localization problem and a baseline. It is possible to see that the baseline has a huge error because it is the simplest prediction model possible. The error obtained by all ML algorithms was lower than 10 cm. The best result was obtained by the **Random Forest** approach, which presents a millimeter error in the distance and 3.05° degrees in orientation. It is interesting to see that the RMSE is higher than the MAE, which indicates that the errors in the predictions do not have the same magnitude, i.e., some predictions can present a small error while others can present a larger error. Through the NRMSE, it is possible to compare the *x* and *y* errors with the θ error, and it demonstrates these are in the same order of magnitude. The $R^{2}$ is always very close to the maximum value (1), showing that the predictions are good. Another important aspect is the training time, which presents a large difference. Random Forest requires a training time that is between 1 and 4 orders of magnitude higher when compared with the other approaches. Furthermore, during the model execution in the robot, the important metric is the response time, for which all the models have good results. The time presented in Table 2 does not include the preprocessing time, which is around 13 milliseconds. It was concluded based on the acquired results that the best approach is the Random Forest Regressor.

Table 3 presents the results for Random Forest but with different grid resolutions during data collection. By analyzing this table, it is possible to see a relationship between the resolution and error. The smaller the grid resolution, the smaller the error obtained in the prediction. This result is interesting because it is possible to see a trade-off: the results can be improved but with the cost of acquiring more images. Another important aspect is that the results were obtained in a small part of the scenario. In the full scenario, these results could be different. Analyzing Figure 12, it is possible to observe this relationship in a visual manner. This relation does not exist on the orientation, because the collection of the data in all cases is at the same rate and performed at 360º degrees for all grid resolutions.

Table 4 shows similar results for the analytical and ML approaches. While the ML presents better performance comparing the MAE error in *x* and *y* (around several millimeters less than the analytical), the orientation error is slightly higher. On the other hand, the analytical is better in the RMSE. This indicates that, in general, the ML approaches have similar results to the analytical approach. It is important to highlight that only the ArUco present on the floor was used, as in [4]. It is also important to note that the results are not intended to reflect the final prediction, but serve as an input for a sensor fusion filter, which will improve the quality of the prediction.

## 5. Conclusions

It is possible to conclude that the results of this work are satisfactory and reached the initial objective. With this, it is possible to validate the initial proposal and state that the use of ML is a good approach to solve the localization problem in RobotAtFactory 4.0. When comparing the ML approach (to predict an estimated pose based on only one ArUco) to the analytical approach, the results are similar. With this, the ML approach is shown to be an interesting approach. Furthermore, using all the ArUcos in the image to predict the pose, the results are also strong (presenting an error in the millimeter scale), showing again that ML is a satisfactory approach to solve the localization problem.

In the analytical approach, the exact pose of each ArUco must be known for the robot to be able to localize itself. On the other hand, in the proposed ML, such explicit knowledge is not necessary. This difference is the main advantage of the approach presented in this work. In scenarios in which the accurate pose of the markers is difficult or impossible to be obtained (for example, in hostile environments), images for training can be collected by a robot that knows its position using another localization system (such as a Differential GPS–DGPS) to create the dataset to train the model. In a later stage, during the localization phase, the DGPS is no longer necessary and accurate localization can be obtained with only a camera.

It is important to note that the results presented in this work are limited because this proposal is not a complete system for localization. In some cases, such as when the robot does not see any ArUcos, the proposed model will not obtain a valid position (because of the ambiguous image problem). To solve this, it is necessary to integrate the present model with other robust approaches, such as an Extended Kalman Filter. Furthermore, another limitation of this work is that the data collection was from a simulator and not a real environment. Thus, to transfer the learning of this work to the real environment, some adjustments must be made (for example, due to the environment’s luminosity).

In future works, it is planned to transfer this model to a real environment, install the model in a robot, and make the necessary adaptations. Furthermore, the creation of a physical module with this ML model embedded is another work proposal. With this module, students and beginners in robotics (especially in RobotAtFactory 4.0) can use this model to solve the localization problem and focus on other tasks, such as planning. Finally, another interesting task is to improve the ML models using GridSearch (https://scikit-learn.org/stable/modules/generated/sklearn.model_selection.GridSearchCV.html, accessed on 17 December 2022) to optimize the parameters. Finally, another proposed direction is to use deep learning approaches to solve the localization problem, such as CNNs and transfer learning (such as VGG16 or GoogLeNet), which can achieve better results.

## Figures and Tables

**Figure 1 sensors-23-03128-f001:**
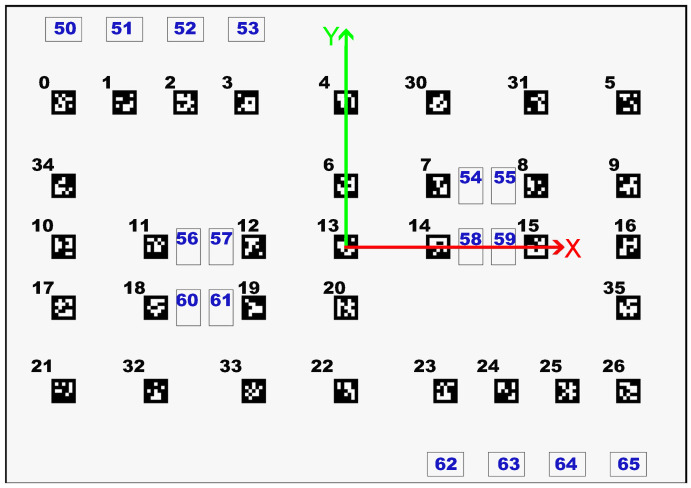
Top view of the RobotAtFactory 4.0 competition field with axes’ representation for identification of each ArUco marker, where black numbers indicate tags on the floor and blue numbers indicate tags on the walls.

**Figure 2 sensors-23-03128-f002:**
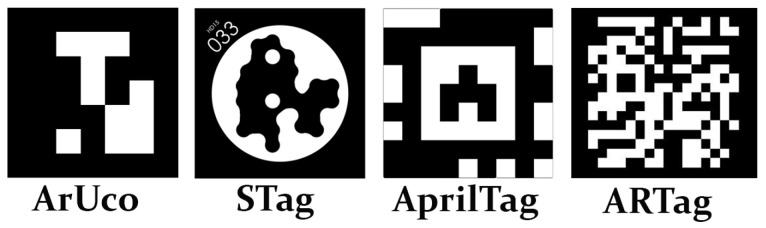
Fiducial marker (Sources to generate: ArUco: https://chev.me/arucogen/, STag: https://github.com/bbenligiray/stag, AprilTag: https://github.com/AprilRobotics/apriltag and ARTag: https://shawnlehner.github.io/ARMaker/, accessed on 15 January 2023) comparison between ArUco Stag, AprilTag, and ARTag [35].

**Figure 3 sensors-23-03128-f003:**
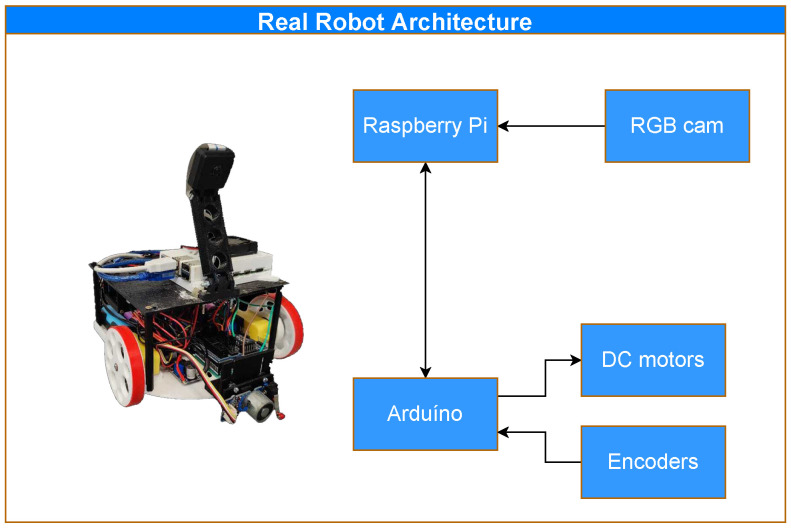
Robot architecture. It is composed of two main micro-controllers: the Raspberry Pi is responsible for the decision-making process and the control of the RGB camera; the Arduino is responsible for the other physical parts, such as the motors and encoders.

**Figure 4 sensors-23-03128-f004:**
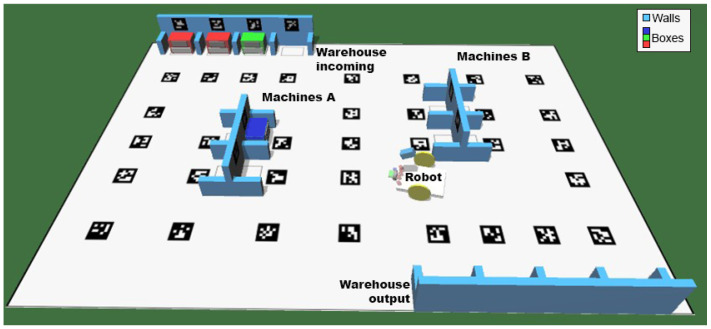
Simulation scene that displays the robot and the environment, where the robot can move and take pictures. The walls, machines, and robot are displayed in the figure with subtitles beside them.

**Figure 5 sensors-23-03128-f005:**
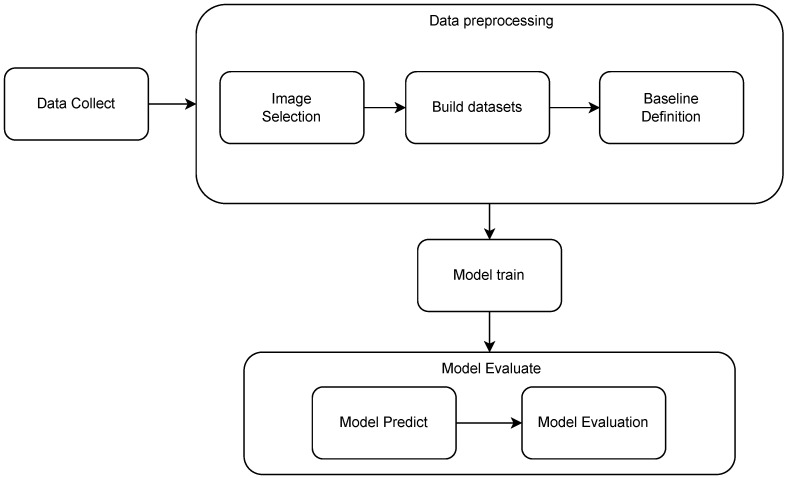
Flowchart of the system. The first step is data collection, followed by data preprocessing. After the data are preprocessed, they are used to train the ML models, and the final step is to evaluate the models, verify their quality, make predictions, and calculate the errors.

**Figure 6 sensors-23-03128-f006:**
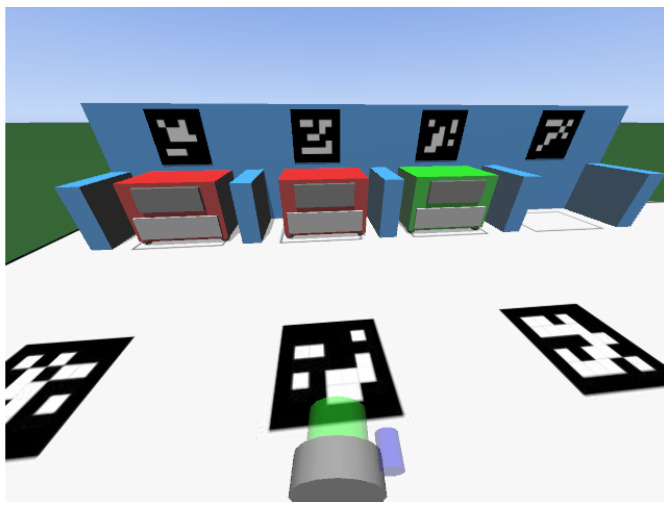
Example of an image taken by the robot’s camera in simulation.

**Figure 7 sensors-23-03128-f007:**
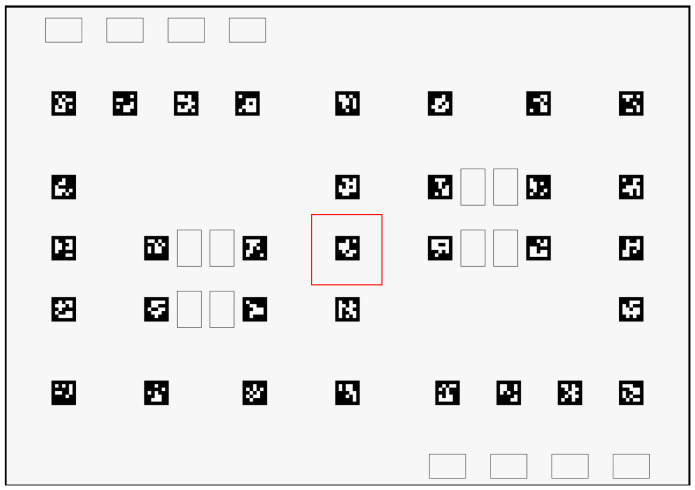
Part of the field used to collect images for the tests to show the relationship between the grid resolution and errors.

**Figure 8 sensors-23-03128-f008:**
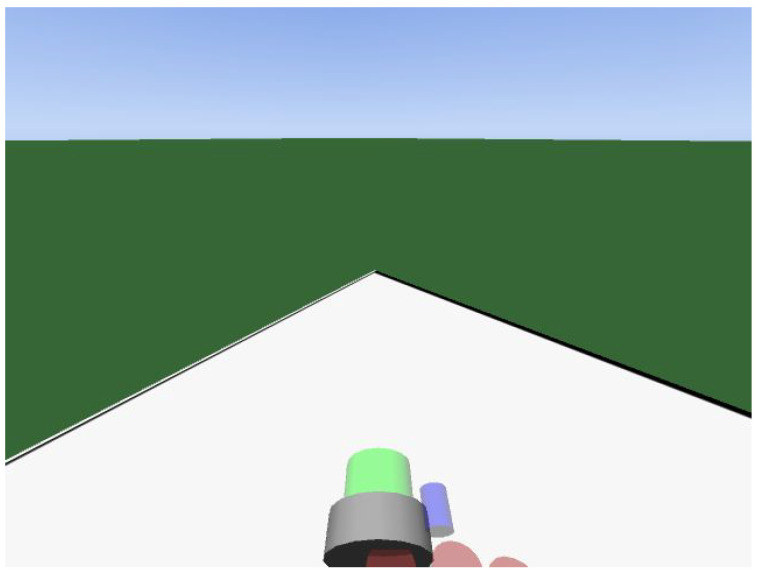
An example of an ambiguous image. In this case, the robot is oriented towards one of the corners of the field. Since no fiducial marker is visible, it is not possible to determine the pose of the robot from the image alone.

**Figure 9 sensors-23-03128-f009:**
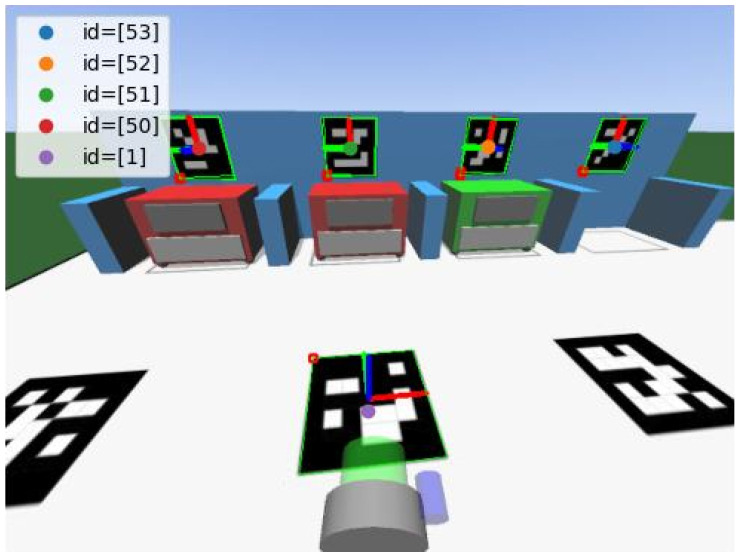
Detected ArUco markers and their corresponding reference frames identified by the OpenCV library.

**Figure 10 sensors-23-03128-f010:**
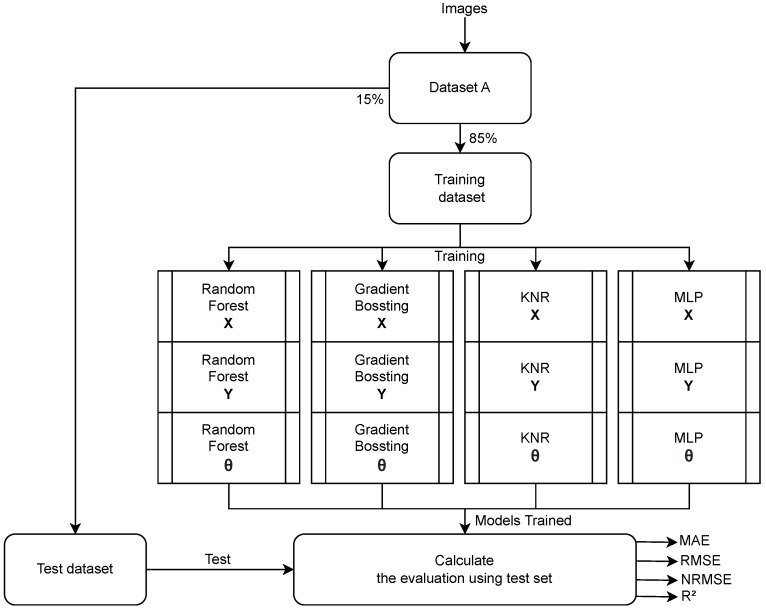
The first process to define the best algorithm. First, dataset A was split into two parts: training and testing. Second, the models were trained: three different models, one for each target variable—*x*, *y* and θ. Finally, the test was executed for each model, and the error measurements were made.

**Figure 11 sensors-23-03128-f011:**
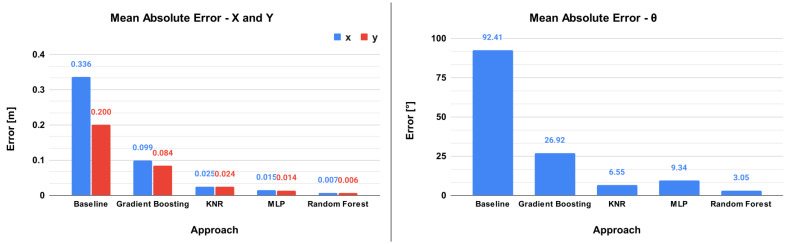
Image on the left displays the MAE for the *x* and *y* axes in meters and the image on the right displays the MAE for the θ in degrees.

**Figure 12 sensors-23-03128-f012:**
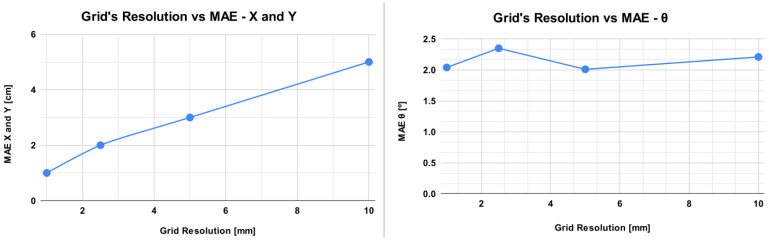
Comparison of the error obtained against the decrease in the grid’s resolution. The graph on the left displays the MAE for the *x* and *y* axes in centimeters (the same curve is for both values), while the image on the right displays the MAE for the θ in degrees.

**Table 1 sensors-23-03128-t001:** ArUco relative pose arrays. The *rvec* values are in the axis–angle format in radians, and *tvec* are in meters.

id	rvec [rad]	tvec [m]
1	2.11, −0.12, 0.08	0.00, 0.02, 0.04
50	−1.89, 1.92, 0.45	−0.06, −0.05, 0.09
51	−1.75, 1.83, 0.46	−0.01, −0.05, 0.10
52	−1.85, 1.92, 0.39	0.04, −0.06, 0.10
53	−1.90, 1.93, 0.41	0.10, −0.06, 0.11

**Table 2 sensors-23-03128-t002:** Results of the proposed approaches ${}^{1}$. The bold values present the best values found.

Approach		Baseline	GB	KNR	MLP	RF
MAE	x [m]	0.336	0.099	0.025	0.015	**0.007**
	y [m]	0.200	0.084	0.024	0.014	**0.006**
	θ [°]	92.41	26.92	6.55	9.34	**3.05**
RMSE	x [m]	0.418	0.133	0.041	0.022	**0.014**
	y [m]	0.226	0.109	0.041	0.021	**0.015**
	θ [°]	105.99	37.85	19.40	21.80	**11.24**
NRMSE	x	0.303	0.096	0.030	0.016	**0.010**
	y	0.310	0.149	0.056	0.030	**0.019**
	θ	0.294	0.105	0.054	0.061	**0.031**
*R*²	x [%]	0.00	0.90	0.99	0.99	**0.99**
	y [%]	0.00	0.77	0.97	0.99	**0.99**
	θ **[%]**	0.00	0.87	0.97	0.96	**0.99**
Training time [s]		-	3302	**28**	1812	15433
Response time avg [ms]		-	**0.08**	4.16	0.12	0.46

^1^ Approaches: GB: Gradient Boosting, KNR: K-Nearest Neighbor Regression, MLP: Multilayer Percepton, RF: Random Forest. Metrics: MAE: Mean Absolute Error, RMSE: Root Mean Squared Error, NRMSE: Normalized Root Mean Squared Error.

**Table 3 sensors-23-03128-t003:** Results ${}^{1}$ using different grid resolutions. The algorithm used to obtain these results was the Random Forest Regressor.

Grid’s Resolution		10 mm	5 mm	2.5 mm	1 mm
Quantity of images		8306	33,281	113,596	655,130
MAE	**x [m]**	0.005	0.003	0.002	0.001
	y [m]	0.005	0.003	0.002	0.001
	θ [°]	2.04	2.35	2.01	2.21
RMSE	x [m]	0.008	0.005	0.003	0.002
	y [m]	0.007	0.005	0.003	0.002
	θ [°]	8.98	12.46	10.35	10.60
NRMSE	x	0.073	0,045	0.030	0.018
	y	0.064	0.043	0.027	0.018
	θ	0.025	0.035	0.029	0.029
*R*²	x [%]	0.95	0.97	0.99	0.99
	y [%]	0.95	0.98	0.99	0.99
	θ [%]	0.99	0.99	0.99	0.99

^1^ Metrics: MAE: Mean Absolute Error, RMSE: Root Mean Squared Error, NRMSE: Normalized Root Mean Squared Error.

**Table 4 sensors-23-03128-t004:** Comparison ${}^{1}$ of the two approaches: analytical and ML (Random Forest). The bold values present the best values found.

Approach		Analytical	ML (Random Forest)
MAE	x [m]	0.028	**0.022**
	y [m]	0.031	**0.023**
	θ [°]	**6.00**	7.27
RMSE	x [m]	**0.034**	0.037
	y [m]	**0.037**	0.043
	θ [°]	**8.89**	14.91
NRMSE	**x**	**0.023**	0.025
	y	**0.044**	0.051
	θ	**0.025**	0.041
Avg percentage Relative error	x [%]	29.6	**27.6**
	y [%]	35.9	**22.2**
	θ [%]	**17.2**	17.3

^1^ Metrics: MAE: Mean Absolute Error, RMSE: Root Mean Squared Error, NRMSE: Normalized Root Mean Squared Error.

## Data Availability

Not applicable.

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
