# Peer review of "A Machine Learning Approach to Robot Localization Using Fiducial Markers in RobotAtFactory 4.0 Competition"

_sensors, 2023, doi:10.3390/s23063128_

Round 1

Reviewer 1 Report

In the paper, the author applies a machine learning method to solve the problem of positioning in the RobotAtFactory 4.0 competition. The airborne camera is used to obtain the relevant data set, and then ML is used to estimate the robot`s position. And the experiment proves that it can obtain the same accuracy as the original location algorithm. We have the following suggestions for further improvement of the paper.

1) It is suggested that the author can use fewer keywords.

2) The format of the article is not uniform. Please check it out more.

3) In the experimental section, we suggest to bold the proposed method and the optimal results.

4) We suggest adding a graph of the localization effect to show the results more clearly.

Author Response

Dear Reviewer,
Thank you for giving us the opportunity to submit a revised final draft of our manuscript entitled "A Machine Learning Approach to Robot Localization using Fiducial Markers in RobotAtFactory 4.0 Competition"  to the Topical Collection Smart Robotics for Automation of  Sensors Journal. We appreciate the time and effort that the reviewer has dedicated to providing your valuable feedback on our manuscript. A thorough revision was performed based on your suggestion and all changes were colored in blue in the revised article.

1) We have removed two keywords (Robot at Factory 4.0 and Autonomous Mobile Robots).

2) We have performed some changes in the format of the article (principally on the image size), and replaced and repositioned the tables, to try to make the article more uniform.

3) We bolded the best results on the table and also we bolded the best approach in the discussion about the results.

4) We also added a graph that shows the error increases with the decrease of the grid’s resolution in the results section.

Reviewer 2 Report

I think emphasizing on "RobotAtFactory 4.0 Competition" in title and keywords it has no truely meanings for this paper, since the method  proposed by the author can be applied maybe to other scenarios, so it is recommended that the author remove "RobotAtFactory 4.0 Competition" from the title and keywords.

Author Response

Dear Reviewer,

Thank you for giving us the opportunity to submit a revised final draft of our manuscript entitled "A Machine Learning Approach to Robot Localization using Fiducial Markers in RobotAtFactory 4.0 Competition" to the Topical Collection Smart Robotics for Automation of  Sensors Journal. We appreciate the time and effort that the reviewer has dedicated to providing your valuable feedback on our manuscript. A thorough revision was performed based on your suggestion and all changes were colored in blue in the revised article.

We acknowledge and thank the reviewer for this valuable comment. A thorough revision was performed based on your comment about the removal of the  "RobotAtFactory 4.0 Competition" from the title and keywords. The authors, after discussions, believe we prefer to maintain this sentence in the title and remove it from the keywords. We consider this because the results of the work are specifically on the competition scenario. We agree with your comment about the possibility to use the method for other contexts, but since the results presented are in a specific context, we believe the sentence should be continuous. Nevertheless, if you don't agree, we are open to changing it according to your comment.

Reviewer 3 Report

The manuscript investigates methods based on artificial intelligence for source localization in the RobotAtFactory 4.0 competition. The work provides an in-depth introduction. Overall, I think the paper needs to undergo major revisions before publication could be considered. My comments, questions and suggestions are listed below.

Introduction: - Authors discuss about source localization using acoustic emission signals, please cite the recent literature about standard threshold and cross-correlation techniques [10.1109/JSEN.2018.2890568, doi.org/10.3390/s20185042].

Methodology: -Did authors used cross-validation in training models?  Cross-validation is used for preventing overfitting, authors should argue which techniques have been applied in their pipeline to handle this problem. The dataset split in training/test is neither sufficient nor cautelative and could represent a lack in content and methodology.

-To compare two items or groups is suggest using also the normalized RMSE (NRMSE).

- How about the computational efficiency of the proposed methods? How about the robustness of the proposed methods against noise effect?

Results: -In results, authors should help readers to better show results with the receiver operating characteristic curve (ROC) and area under the ROC curve (AUC) for each comparison.

My final suggestion is to check the English, as it appears evident that authors are not English native speakers.

Author Response

Dear Reviewer,
Thank you for giving us the opportunity to submit a revised final draft of our manuscript entitled "A Machine Learning Approach to Robot Localization using Fiducial Markers in RobotAtFactory 4.0 Competition" to the Topical Collection Smart Robotics for Automation of  Sensors Journal. We appreciate the time and effort that the reviewer has dedicated to providing your valuable feedback on our manuscript. A thorough revision was performed and all changes were colored in blue in the revised article.

1) We acknowledge and thank you for the indication of the article about localization using acoustic emission signals [10.1109/JSEN.2018.2890568, doi.org/10.3390/s20185042]. We performed a modification in our text and now this interesting article is cited in the state of the art.

2) We acknowledge and thank the reviewer for this valuable comment. We understand the purpose and importance of cross-validation, and this was a topic of discussion within the group prior to submitting the document for review. Finally, the decision was made not to use cross-validation since this work intends to demonstrate the possibility of using machine learning to solve the localization problem. As it is not a finished version at all, there is still a lot of work to be done to get there, namely increasing the data set, optimizing the algorithms, and carrying out more tests. Finding it makes more sense to use cross-validation in this next stage of the work. Another reason that led us to decide not to use cross-validation at this time was the size of the dataset used (300,000 training images and 45,000 test images), as well as all the pre-processing carried out, recognizing the possibility of overfitting, we also recognized that it decreases with the increase in the dataset size. Lastly, emphasizing the sensitivity acquired during the training carried out, it was possible to perceive the constant behavior of the results even though they were not exposed in the document since different proportions of training and testing were used. Finally, we recognize once again the importance of cross-validation and agree with the reviewer on this topic, however, supporting our decision on the arguments made available, we ask you to consider the not usage of that at this moment.

3) We acknowledge and thank the reviewer for this valuable comment about the computational efficiency and their robustness against noise. We agree these are important discussions about the methods. These topics are planned to be discussed in-depth in future works when these methods will be used in a real scenario. Since a simulator was used, these aspects can not be well discussed when compared with the real world. So, in this way, these topics will be focused on in detail in future works. 

4) Regarding the Receiver operating characteristic curve (ROC) and area under the ROC curve (AUC), according to our research, this kind of graph is for classification problems, but in our case, a regression was applied. Anyway, based on your good suggestion, we added a graph that shows the evolution of the error with the decrease of the grid’s resolution in the results section, to help the reader understand the relationship between error and resolution.

5) About the NRSME,  we have added this metric to the article and it was possible to compare the different groups with different unit measurements. We added this metric in all tables.

6) Thank you for your suggestion about the English review. We considered this carefully, and performed some changes in the text, fixing some words and conjugation of verbs.

Round 2

Reviewer 3 Report

The authors have satisfactorily addressed my concerns.